# Glycosylation and Characterization of Human Transferrin in an End-Stage Kidney Disease

**DOI:** 10.3390/ijms25094625

**Published:** 2024-04-24

**Authors:** Goran Miljuš, Ana Penezić, Lucia Pažitná, Nikola Gligorijević, Marko Baralić, Aleksandra Vilotić, Miloš Šunderić, Dragana Robajac, Zorana Dobrijević, Jaroslav Katrlík, Olgica Nedić

**Affiliations:** 1Institute for the Application of Nuclear Energy (INEP), Department for Metabolism, University of Belgrade, 11000 Belgrade, Serbia; 2Institute of Chemistry, Slovak Academy of Sciences, 84538 Bratislava, Slovakia; 3Institute of Chemistry, Technology and Metallurgy, Department of Chemistry, University of Belgrade, 11000 Belgrade, Serbia; 4School of Medicine, University of Belgrade, 11000 Belgrade, Serbia; 5Clinic of Nephrology, University Clinical Centre of Serbia, 11000 Belgrade, Serbia; 6Institute for the Application of Nuclear Energy (INEP), Department for Biology of Reproduction, University of Belgrade, 11000 Belgrade, Serbia

**Keywords:** transferrin, glycosylation, end-stage kidney disease, peritoneal dialysis, structure, lectin microarray, FTIR

## Abstract

Chronic kidney disease (CKD) is a global health concern affecting approximately one billion individuals worldwide. End-stage kidney disease (ESKD), the most severe form of CKD, is often accompanied by anemia. Peritoneal dialysis (PD), a common treatment for ESKD, utilizes the peritoneum for solute transfer but is associated with complications including protein loss, including transferrin (Tf) a key protein involved in iron transport. This study investigated Tf characteristics in ESKD patients compared to healthy individuals using lectin microarray, spectroscopic techniques and immunocytochemical analysis to assess Tf interaction with transferrin receptors (TfRs). ESKD patients exhibited altered Tf glycosylation patterns, evidenced by significant changes in lectin reactivity compared to healthy controls. However, structural analyses revealed no significant differences in the Tf secondary or tertiary structures between the two groups. A functional analysis demonstrated comparable Tf-TfR interaction in both PD and healthy samples. Despite significant alterations in Tf glycosylation, structural integrity and Tf-TfR interaction remained preserved in PD patients. These findings suggest that while glycosylation changes may influence iron metabolism, they do not impair Tf function. The study highlights the importance of a glucose-free dialysis solutions in managing anemia exacerbation in PD patients with poorly controlled anemia, potentially offering a targeted therapeutic approach to improve patient outcomes.

## 1. Introduction

Chronic kidney disease (CKD) is a condition characterized by the impaired function of the kidneys, which are responsible for the regulation of electrolyte concentrations, extracellular fluid volume, blood pressure and hormone secretion [1]. CKD is also characterized by anemia, which can additionally aggravate CKD itself, especially in patients with diabetes mellitus who already have various cardiovascular complications [2]. It is estimated that up to 13.4% of the global population is affected by CKD, which is approximately one billion people worldwide [3,4]. The treatment of CKD is an enormous economic burden, and the mortality rate is high [5,6,7]. The severity of CKD is classified into five stages. Stage 5, with a glomerular filtration rate of less than 15 mL/min/1.73 m^2^, implies that the kidneys are no longer capable of efficient blood filtration and that replacement therapy is needed [8]. This stage of CKD is also defined as End-Stage Kidney Disease (ESKD). Patients with ESKD are at high risk of fatal outcomes caused by amplified cardiovascular and other complications, some of which can be worsened by anemia [9]. New strategies for the early detection of CKD, efficient treatment and reduced advancement to ESKD, relying on a better understanding of the pathological mechanisms involved, are needed.

Peritoneal dialysis (PD), which is applied for the treatment of ESKD, is a procedure where the peritoneum is used as a natural membrane for the transfer of blood solutes and is applied with approximately 11% of patients with ESKD requiring dialysis treatment [10]. The method is based on the introduction of hyperosmolar solution to the abdomen and is associated with many health risks. The major problem in PD may arise from the altered role of the peritoneum, which produces pro-inflammatory cytokines that stimulate neovascularization, the infiltration of immune cells and changes in the structure and permeability of the peritoneum [11]. Consequently, substantial protein loss may occur (estimated to be 5–15 g per day), which may impair the maintenance of the osmotic equilibrium and the efficiency of PD [12]. The extent of protein “leakage” depends on the protein itself, mostly its molecular mass. Transferrin is one of the proteins affected by ESKD and PD. 

Transferrin (Tf, 79 kDa) concentration in patients on PD is reduced, consequently causing anemia, since Tf is a transporter of iron ions. N- and C-termini of Tf are globular lobes that shape two high-affinity iron-binding sites. This protein is glycosylated and exists in several glycoforms [13,14,15]. Tf enters cells via Tf receptors (TfR1 and/or TfR2), and the principal cells that bind Tf are erythroid cells, macrophages, hepatocytes, intestinal and placental cells [16,17,18]. Both the concentration and the structure of Tf are important for the proper metabolism of iron and associated processes. Glycosylation and oxidation influence protein conformation and/or the position of interacting residues possibly affecting its interaction with ions, ligands, receptors, proteases or other binding partners. Pathological conditions may result in an irregular functioning of cellular glycosylation machinery, as was confirmed through Tf glycosylation analysis in liver diseases [19,20]. Furthermore, ESKD is accompanied by oxidative stress, which is known to generate free radicals that modify proteins [21]. Having in mind that patients with ESKD subjected to PD have lower concentrations of Tf and iron in their circulation, often suffering from severe anemia, which requires targeted therapy, it seemed relevant to investigate whether the Tf molecule itself undergoes structural changes in this health condition, which may influence Tf function. Lectin-based microarray and spectroscopic techniques were employed to compare the structure of Tf isolated from healthy persons and patients on PD, while Tf interaction with TfR was examined via immunocytochemistry using placental cells.

## 2. Results

### 2.1. Hematological and Biochemical Parameters in Patients on Peritoneal Dialysis

Blood parameters monitoring the status of iron metabolism and associated molecules/processes in PD patients are listed in Table 1, together with reference ranges obtained from datasheets accompanying the reagents. Medians for the majority of the parameters are at or below the lower reference limit except for ferritin, which was, expectedly, close to the upper reference limit. The obtained data point to anemia and protein loss. The concentration of Tf is in accordance with other parameters—at the lower edge of the reference range.

Tf was isolated from each individual serum sample and used to study its glycan structure, using a lectin microarray, as well as its secondary and tertiary structures using several spectroscopic methods. An immunocytochemical analysis was performed in order to assess the possible effect of a disease on the reactivity of this molecule, i.e., its interaction with TfR.

### 2.2. Glycosylation of PD Patients Transferrin

PD transferrin from patients was shown to interact with the majority of lectins used in the microarray, with S/N > 3, except for MAL-I and -II (Table 2). Taking into account the glycan specificity of lectins that interacted with Tf, a wide array of glycan structures was present on Tf. When the criterion for strong reactivity was applied (i.e., fluorescent intensity > 250 AU), nine lectins remained in the category of strong binders.

In data processing, intensities of signals derived from all 15 lectin interactions originating from one sample were summed up and termed 100% (for each sample). Individual reactivity for each lectin was expressed as a portion in the total lectin response (%). A statistical analysis of the results for the two study groups showed significant differences in the reactivity of Tf with seven lectins (Figure 1).

Transferrin molecules originating from PD patients expressed higher reactivity with lectins specific for core Fuc (AAL and PhoSL), β-1,4GlcNAc dimers and oligomers (DSL and WGA) and GlcNAc/αMan with core Fuc (LCA), and expressed reduced reactivity with those specific for αMan (ConA) and Gal (RCA). Thus, the glycosylation pattern of Tf changes significantly in patients with ESKD on PD.

### 2.3. Secondary and Tertiary Structures of PD Transferrin

An FTIR analysis of the isolated Tf from two study groups (aggregate samples) was used to assess whether there are differences with respect to the secondary structure of Tf (Figure 2). No spectral shifts, significant differences in the intensity of specific peaks or relative portions of specific secondary structures were detected between the two sample groups, although the portion of turns was increased, from approximately 9% in healthy to 14% in PD at the expense of an α-helix structure, which was reduced from approximately 44% in healthy persons to 38% in patients on PD. Absorbance signals were baseline corrected, and a second derivative was calculated and is presented on the graph.

Fluorescence emission spectra of the isolated Tf from the two groups also showed no significant difference regarding the peak position or its intensity, confirming the absence of alterations in the tertiary structure of Tf in PD patients (Figure 3).

Additionally, a spectrophotometric analysis of the environment surrounding aromatic amino acid residues [22] in isolated Tf molecules from two groups was performed by recording spectra in the range 270–300 nm, derivatizing signal intensities to the fourth order and processing them using the Savitzky–Golay smoothing method (Figure 4). No significant difference was observed between two groups of Tf samples, although a shift of less than 1 nm in PD samples was recorded.

### 2.4. Functional Analysis of PD Transferrin

A functional analysis was performed using the treatment of HTR-8/SVneo cells with isolated Tf from two groups of samples. Tf interacts with cell surface TfRs, and the complex becomes internalized. The idea was to evaluate if a PD Tf interacts with its receptor or this interaction is not taking place due to altered Tf molecules in PD patients. An immunocytochemical analysis with a simultaneous application of anti-Tf and anti-TfR antibodies to Tf-treated HTR-8/SVneo cells revealed that Tf from two groups of samples equally interacted with the TfR (Figure 5).

## 3. Discussion

Peritoneal dialysis is one of the three complementary methods of kidney replacement therapy (KRT), which are used to treat patients with ESKD [23]. The most common associated disease is anemia in CKD, which requires the use of recombinant erythropoiesis-stimulating agents (ESAs) in most dialysis patients [24]. Before starting the prescription of ESAs according to KDIQO recommendations, it is necessary to rule out acute inflammation, regulate secondary hyperparathyroidism (SHPT) and replace iron depots [23,24], which was carried out in our cohort. The assessment of feremia in PD patients is performed on the basis of serum Fe, TIBC and ferritin [25]. Substitution was carried out by administering heme and nonheme Fe orally, considering that PD is a form of home treatment [24,25]. Given that the level of Fe absorption in the duodenum is affected by the various drugs that this group of patients takes (proton pump inhibitors, phosphate binders), as well as the edema of the intestinal mucosa due to hypervolemia [26], it is necessary to maintain the patients’ dry body weight, which was fulfilled in our group; the patients were euvolemic, and according to the parameters of dialysis, they underwent a competent method of KRT, which corresponds to the literature data [27,28]. 

Transferrin exists in three forms of varying abundance in the circulation: apo-Tf (without Fe ions), mono-Tf (with one Fe ion) and holo-Tf (saturated with two Fe ions). Tf has N- and C-lobes, both further divided into two subdomains linked by a short binding region. Each of the lobes contains one iron-binding site. The C-lobe is N-glycosylated at positions 413 and 611 (corresponding to Asn432 and Asn630 of the full protein sequence), which can be variously occupied, resulting in a number of glycoforms. The most prevalent Tf glycoform contains two bi-antennary glycans with four sialic acid (Sia) residues (tetrasialo-Tf), while minor glycoforms contain two, three or five Sia residues, some of which are α1,6 fucosylated at the chitobiosyl core [13,14,15]. According to Ghanbari et al. in their molecular dynamic simulation analysis, glycosylation appears to affect the layout of the Fe-binding site residue and transferrin function [29]. Under physiological conditions, 30% of Tf, on average, is saturated with iron, while residual molecules buffer the free iron ion concentration. Tf saturation often falls below 20% in ESKD patients [30,31,32]. A loss or decrease in Tf sialylation affects its catabolism in liver, as the protein with galactose capping interacts with the receptor more easily, leading to its faster clearance and shorter lifespan [33,34]. The microarray results presented here show a lower response of Tf from PD samples with galactose-specific lectin (RCA), so we can speculate on its faster removal from blood and further disturbances in iron metabolism contributing to, e.g., anemia. In addition, anemia in ESKD can be caused by decreased erythropoietin production and an increased rate of erythrocyte clearance as well [35,36]. 

The results obtained in this study confirm low iron and Tf concentrations in the serum of ESKD patients on PD and document a significantly altered glycosylation of Tf. In general, the change in glycosylation may arise from alterations in the expression and/or activity of different transferases and glycosidases, as well as the availability of sugar substrates [37]. Mass spectrometry is regarded as a “gold standard” in the determination of glycan structures [38,39], and no decisive conclusions can be made without this method. It is not usually used as a high-throughput approach that allows comparisons between large groups of samples. Thus, we used a lectin microarray method, which is a technique generally applied for high-throughput glycoprofile screening. Our screening platform showed statistically relevant differences in the reactivity of PD transferrin molecules with the specific lectins (AAL, PhoSL, DSL, WGA, LCA, ConA and RCA) compared to controls, indicating a possibility of an aberrant Tf glycosylation pattern in ESKD. However, these differences had no overall effect on the conformation of the Tf molecule. The interaction of PD Tf with the Tf receptor took place, showing that the differences in the glycosylation pattern do not affect the internalization of PD Tf. These data suggest that saccharides do not seem to be influential for Tf/TfR recognition and interaction per se, although the results are inconclusive with respect to the affinity and kinetic parameters of this interaction. On the other hand, the altered glycosylation of Tf may possibly influence iron ion binding or clearance rate and path, as documented for some other glycoproteins [40]. This study represents the first experimental work on possible interconnections between the Tf glycosylation pattern and iron transport in kidney disease. Our results suggest that the effect of altered glycosylation might have a huge influence on iron binding/release to and from Tf regarding anemia in PD patients. These results warrant further in-depth structural studies and validation with more specific methods. 

Common practice for the dialysis treatment of ESKD patients is the use of dialysates with glucose concentrations up to a 100 mmol/L (most commonly 5.5 mmol/L), due to the prevention of hypoglycemia and hypotension [41]. The use of hyperosmolar glucose dialysis solutions is a source of carbohydrates and glycating agents. Considering that an altered glycosylation and/or glycation of transferrin affects the metabolism of Fe, thus worsening anemia, the use of exclusively glucose-free dialysis solutions might be the therapy of choice for PD patients with poorly regulated anemia, which requires high doses of ESAs. The results of our study strongly suggest that special attention should be paid to these patients, potentially providing a targeted therapeutic approach to improve patient prognosis. 

## 4. Materials and Methods

### 4.1. Blood Samples

Blood samples from patients on PD were obtained from the University Clinical Centre of Serbia (n = 75, 31 males, 44 females, age: 32–87 years) together with samples from healthy persons (n = 30, 13 males, 17 females, age: 32–87 years). Patients were diagnosed with CKD stage 5 and were under PD for at least six months, four times a day with 2–3 L of PD fluid. The appropriateness of the dialysis was confirmed via the mean Kt/V, which was 2.33 ± 0.49. Patients diagnosed with anemia (59 patients, Hb ≤ 110 g/L) were treated according to KDIGO guidelines, which included the supplementation of erythropoietin-stimulating agents (ESAs, injection) and iron (per os). Since transferrin is synthesized in the liver, the inclusion criteria for patients entering the study were the following: the synthetic function of the liver preserved, hepatotropic viruses being negative and stable liver enzymes. The study was approved by the Ethical Committees of the University Clinical Centre of Serbia (approval number: 890/8) and INEP (approval number: 02-903/3) and was carried out in accordance with the Declaration of Helsinki and Ethical guidelines for medical and health research involving human subjects. Informed consent was obtained from all participants.

Blood samples (with and without anticoagulant) were collected after overnight fasting, and before dialysis in the case of patients. Whole blood was used to determine hematological parameters, while serum was used to measure biochemical analytes and for Tf extraction. Laboratory measurements were performed using standard procedures, commercial assays and clinical analyzers. Results were expressed as medians and percentile ranks (2.5th and 97.5th) since some parameters showed deviations from normal distributions.

### 4.2. Isolation of Transferrin

Transferrin was isolated from each serum sample separately through a three-step procedure [42]. In brief, serum (0.5 mL) was mixed with 1.2% rivanol solution (1:1, *v*:*v*) and subjected to centrifugation at 13,000× *g* for 10 min. A precipitate was discarded, and the procedure step was repeated. The supernatant was passed through a charcoal column (which retained rivanol), and the eluate was mixed with the saturated ammonium sulfate solution (SASS) to reach 50% saturation. After centrifugation at 13,000× *g* for 10 min, the precipitate was discarded, and the supernatant was mixed with SASS to reach 70% saturation. The centrifugation was repeated, and the resulting supernatant contained Tf of high purity, which was used in further experiments. Isolated Tf was stored at −80 °C until used for further analysis.

### 4.3. Lectin-Based Microarray Analysis of Transferrin

A microarray analysis was performed with each sample separately using biotinylated lectins, epoxy microarray slides (NEXTERION Slide E, Schott, Germany), a non-contact piezoelectric sciFLEXARRAYER S1 microarray spotter and a piezo dispense capillary PDC 80 (Scienion AG, Berlin, Germany) [43]. In brief, Tf samples (0.1 mg/mL, 1.2 nL per spot, in triplicate) were printed on microarray slides in the form of 8 identical subarrays, and slides with printed Tf samples were left at room temperature and a humidity of 65% overnight for sample immobilization. Eight-well microarray slide masks (Grace Bio-Labs, Bend, OR, USA) were applied on slides and free epoxy groups blocked with 3% BSA for 1 h. Slides were washed with PBS containing 0.1% Tween-20 (PBST), and 15 biotinylated lectins (25 μg/mL in PBST) with different glyco-specificity to human glycoproteins were loaded into slide mask wells and left at room temperature for 1 h. All lectins were purchased from Vector Laboratories (Burlingame, CA, USA) except recombinant PhoSL, which was a kind gift from Dr. Seonghun Kim (Korea Research Institute of Bioscience and Biotechnology (KRIBB), Jeonbuk Branch Institute, Jeongeup, Republic of Korea). Slides were washed again and incubated with streptavidin conjugated with a fluorescent dye CF647 (0.5 µg/mL in PBST, Biotium, Hayward, CA, USA) at room temperature in the dark for 15 min. After the last washing and drying step, fluorescent signals were measured using an InnoScan^®^710 fluorescent microarray scanner (Innopsys, Carbonne, France) at the wavelength of 635 nm.

The intensity values of the signals were expressed in arbitrary units (AUs), and when it was >250 AU, the interaction was considered to be strong. Lectins used for this analysis are presented together with the results (Table 2). Each sample was assayed in triplicate, and specific signals (S) were corrected for the background “noise” signal (N). Signals were analyzed using Mapix^®^ 7.4.1 software (Innopsys). The cut-off value for the specificity was set at S/N > 3. The results for two study groups were expressed as means ± SD, and the difference between groups was determined using a Student’s *t*-test (at *p* < 0.05).

### 4.4. Fourier-Transform Infrared Spectroscopic (FTIR) Analysis of Transferrin

Individual Tf samples from two study groups were pooled into two aggregate samples for all spectral analyses: from healthy persons and PD patients. Aggregate samples were prepared by mixing 10 µL of each isolated Tf (2 mg/mL or 25 µmol/L) belonging to the same study group. Samples were placed on ZnSe windows (32 mm × 2 mm, Thermo, Madison, MA, USA) and dried under a stream of nitrogen, and FTIR measurements were performed on an IRAffinity-1 FTIR spectrophotometer (Schimadzu, Kyoto, Japan). Spectra were acquired in transmission mode (100 scans per sample) with a resolution of 4 cm^−1^. Origin Pro 8.6 software (OriginLab, Northampton, MA, USA) was used to process the spectra (smoothing and baseline corrections) [44]. The deconvolution of the amide I peak was performed also using Origin Pro 8.6 software. Each spectrum was recorded in triplicate and corrected for the spectrum obtained for PBS.

### 4.5. Fluorescence Emission Spectroscopic Analysis of Transferrin

Fluorescence emission spectra were recorded for aggregate Tf samples (0.2 µmol/L) on an RF6000 spectrofluorimeter (Shimadzu, Kyoto, Japan) in the wavelength range 310–420 nm following excitation at 295 nm, using quartz cell (1 cm path length) and slit widths of 4 nm. Each spectrum was recorded in triplicate and corrected for the spectrum obtained for PBS.

### 4.6. UV-VIS Spectrophotometric Analysis of Transferrin

UV spectra of aggregate Tf samples (1 mg/mL or 12.5 µmol/L) were recorded on the spectrophotometer UV-1900i (Shimadzu, Kyoto, Japan), using quartz cuvette (1 cm). Each spectrum was recorded in the range 270–300 nm, in triplicate, and corrected for the spectrum of PBS. The spectra were recorded with a bandwidth of 1 nm, data interval of 0.1 nm and scanning speed of 2 nm/min. The obtained signals were derivatized to the 4th order and further processed using the Savitzky–Golay smoothing method with a 150-point window. Origin Pro 8.6 software (OriginLab, Northampton, MA, USA) was used for signal processing.

### 4.7. Functional Analysis of Transferrin via Immunocytochemistry

Human extravillous trophoblast HTR-8/SVneo cells were seeded in a 24-well plate on glass coverslips (1 × 10^5^ cells/well) in RPMI 1640 culture medium (Gibco, Paisley, UK) supplemented with 10% heat-inactivated fetal calf serum (*v*:*v*, FCS, Sigma Aldrich, St. Louis, MI, USA) and 1% antibiotic/antimycotic solution (Capricorn Scientific, Ebsdorfergrund, Germany) overnight. After rinsing with PBS, the cells were incubated in serum-free culture medium for 2 h to remove Tf from FCS. Subsequently, the cells were treated with Tf isolated from serum samples originating from the two study groups: patients on PD and healthy persons (aggregate samples, 100 µg/mL) in serum free culture medium for 2 h. Control cells were kept in serum-free culture medium only. After incubation, cells were rinsed with PBS, fixed in 4% paraformaldehyde (at room temperature for 15 min), permeabilized with 0.1% of Triton X-100 in PBS (at room temperature for 10 min) and further incubated with 1% BSA in PBS/0.05% Tween-20 at room temperature for 1 h to disable the non-specific binding of antibodies. Simultaneous incubation with primary goat anti-human Tf (1:50, INEP, Belgrade, Serbia) and mouse anti-human TfR (CD71, 1:100, Santa Cruz Biotechnology Inc., Santa Cruz, CA, USA) antibodies took place in a humidified chamber at 4 °C overnight. The visualization of the primary antibody binding was performed using anti-goat IgG Alexa 488 and anti-mouse IgG Alexa 555 (both 1:1000, Invitrogen, Carlsbad, CA, USA) antibodies, respectively, which were incubated with samples at room temperature for 1 h. Non-specific binding controls were prepared without primary antibodies. Nuclei were counterstained with Vectashield Mounting Medium with DAPI (Vector Laboratories, Burlingame, CA, USA). Images were taken using a 40× objective lens on a Carl Zeiss Axio Imager. A1 microscope with an AxioCam MRm camera (Carl Zeiss, Jena, Germany).

## Figures and Tables

**Figure 1 ijms-25-04625-f001:**
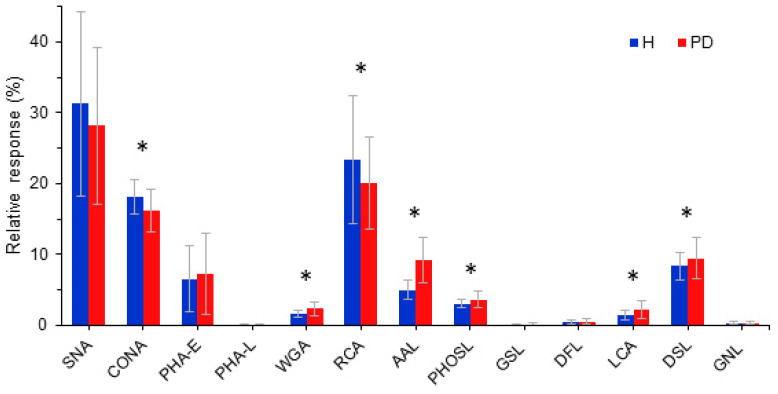
Reactivity of transferrin isolated from healthy persons (H) and patients on peritoneal dialysis (PD) with lectins. A statistically significant difference between groups (*p* < 0.05) is labeled with “*”.

**Figure 2 ijms-25-04625-f002:**
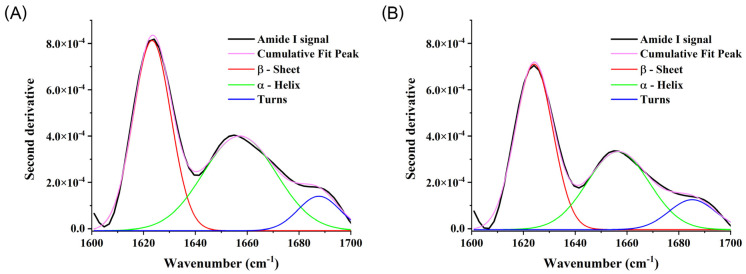
FTIR spectra of the isolated transferrin from healthy persons (**A**) and patients on PD (**B**). The absorbance signal is presented as a second derivative.

**Figure 3 ijms-25-04625-f003:**
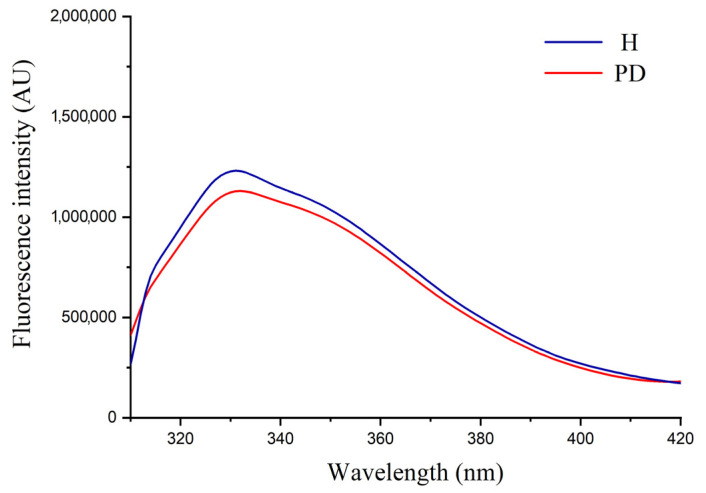
Fluorescence emission spectra of the isolated transferrin from healthy persons (H) and patients on peritoneal dialysis (PD).

**Figure 4 ijms-25-04625-f004:**
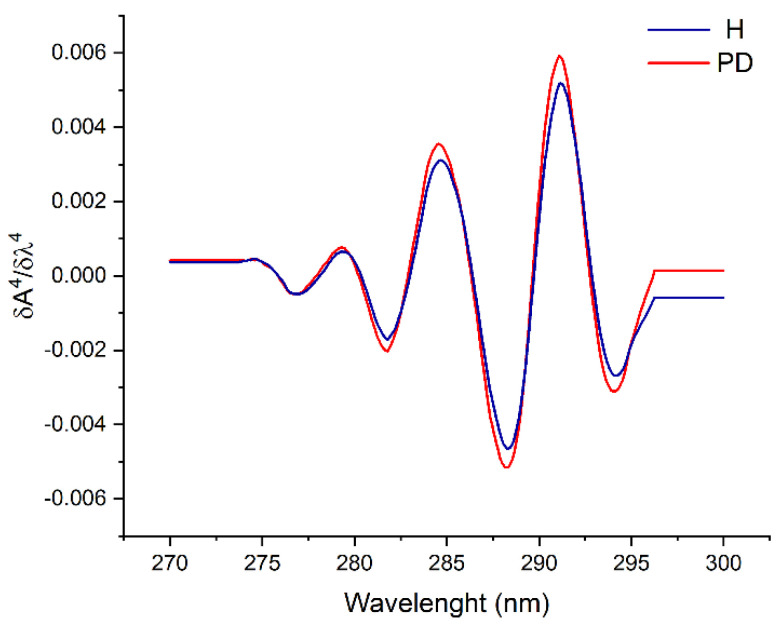
UV spectra of the isolated transferrin from healthy persons (H) and patients on peritoneal dialysis (PD).

**Figure 5 ijms-25-04625-f005:**
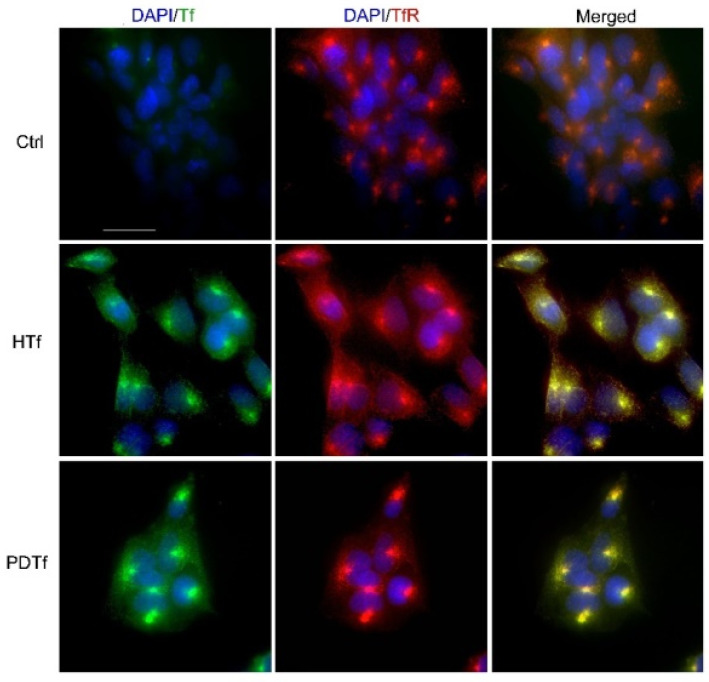
Immunocytochemical staining of HTR-8/Svneo cells using anti-Tf (DAPI/Tf, blue/green) and anti-TfR (DAPI/TfR, blue/red) antibodies. Cells were grown in serum-free conditions (Ctrl) or in a medium supplemented with Tf from healthy persons (HTf) or patients on peritoneal dialysis (PDTf). The co-localization of Tf and TfR in both treatments is shown in the merged images.

**Table 1 ijms-25-04625-t001:** Concentrations of biochemical and hematological parameters related to iron metabolism in patients on peritoneal dialysis (PD), expressed as medians and 2.5–97.5th percentile rank.

Concentration	Patients on PD (n = 75)	Reference Range
Erythrocytes (×10^12^/L)	3.4 (0.3–4.3)	4.3–5.7
Hemoglobin (g/L)	103 (72–129)	138–175
Hematocrit (L/L)	0.32 (0.22–0.41)	0.41–0.53
Total protein (g/L)	64.0 (54.9–77.0)	62.0–81.0
Albumin (g/L)	37.0 (28.0–44.1)	35.0–53.0
TIBC (µmol/L)	44.7 (27.2–57.8)	44.8–75.1
Transferrin (g/L)	1.7 (0.9–2.6)	1.7–3.8
Ferritin (µg/L)	227 (47–1587)	30–400
Iron (µmol/L)	11.5 (7.6–19.7)	11.0–30.0
Transferrin saturation (%)	28.0 (17.9–49.3)	20.0–50.0

**Table 2 ijms-25-04625-t002:** Lectins used in microarray; their glycan specificity and reactivity with a PD transferrin (weak “+” and strong “++”).

Lectin (Source)	Glycan Specificity	Reactivity with Tf
SNA (*Sambucus nigra*)	NeuNAcα2,6Gal/GalNAc	++
ConA (*Conavalia ensiformis*)	Manα1,6(Manα1,3)Man	++
MAL-I (*Maackia amurensis*)	NeuNAcα2,3Galβ1,4GlcNAc	-
MAL-II (*Maackia amurensis*)	NeuNAcα2,3Galβ1,3(±NeuNAc2,6)GalNAc	-
PHA-E (*Phaseolus vulgaris*)	Galβ1,4GlcNAcβ1,2Man with bisecting GlcNAc	++
PHA-L (*Phaseolus vulgaris*)	Tri/tetraantennary complex type N-glycans w/terminal Gal	+
WGA (*Triticum vulgaris*)	GlcNAcβ1,4GlcNAc; chitin oligomers; NeuAc	++
RCA (*Ricinus communis*)	Galβ1,4GlcNAc	++
AAL (*Aleuria aurantia*)	Fucα1,6GlcNAc; Fucα1,3(Galβ1,4)GlcNAc	++
PHOSL (*Pholiota squarrosa*)	Fucα1,6GlcNAc	++
GSL-I (*Griffonia simplicifolia*)	Galα1,3Gal; Galα1,3GalNAc	+
LCA (*Lens culinaris*)	αDGlc, αDMan in N-glycans with Fuca1,6GlcNAc	++
DSL (*Datura stramonium*)	GlcNAcβ1,4GlcNAc oligomers; Galβ1,4GlcNAc	++
DFL (*Narcissus pseudonarcissus*)	α-linked mannose, preferring polymannose structures containing (α-1,6) linkages	+
GNL (*Galanthus nivalis*)	structures containing (α-1,3) mannose residues	+

## Data Availability

The data supporting the findings of this study are available from the corresponding author upon reasonable request.

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
