# Peer review of "Glycosylation and Characterization of Human Transferrin in an End-Stage Kidney Disease"

_ijms, 2024, doi:10.3390/ijms25094625_

Round 1

Reviewer 1 Report

Comments and Suggestions for Authors

Thank you for letting me review this article named "Glycosylation and characterisation of human transferrin in an end stage kidney disease" by Goran Miljuš et al.

As of now there have been many studies on human serum transferrin glycosylation, mainly around congenital glycosylation disorders, alcohol abuse, and hepatobiliary disorders such as hepatocellular carcinoma. Rarely reported in chronic kidney disease. Thus, the topic chosen for this work is novel and clinically relevant.

The authors chose appropriate subjects for the study and designed a sound research protocol and experimental methodology. The authors investigated transferrin (Tf) characteristics in ESKD patients compared to healthy individuals using lectin microarray, spectroscopic techniques and immunocytochemical analysis to assess Tf interaction with transferrin receptors (TfR).  They found altered patterns of Tf glycosylation in ESKD patients compared to healthy controls, with no significant differences in Tf secondary or tertiary structure between the two groups. Simultaneous functional analyses showed that structural integrity and Tf-TfR interactions remained unchanged in patients with PD despite significant alterations in Tf glycosylation. These results suggest that although glycosylation changes may affect iron metabolism, they do not impair Tf function.

My questions and suggestions to improve the overall quality of the manuscript are given below:

1. In discussion section, it has been claimed that “the use of exclusively glucose-free dialysis solutions might be the therapy of choice in PD patients with poorly regulated anemia which requires high doses of ESA.” Currently, sugar-containing dialysate with a glucose concentration of 5.5 mmol/L is mostly used in maintenance haemodialysis patients, which has been shown in several studies to reduce the incidence of hypoglycaemia and hypotension during dialysis, and does not affect the indicators of serum albumin, glycated haemoglobin and lipids, which is safe and reliable. Therefore, further consideration should be given to the authors' statement regarding the selection of dialysate components for this group of patients. The significance of this paper has not been fully elaborated. The authors need to emphasise the innovative contribution of this paper.

2. Line no. 160: “…the use of recombinant ESA in most…”could refine the full name of the ESA (erythropoiesis-stimulating agent).

3. Finally,I suggest only a minor revision of the English language for some grammatical errors and a check on the references.

4. There are some problems about typesetting or format of the paper. In the part of introduction, it is more clear that describing the links between transferrin and peritoneal dialysis in end stage kidney disease in other paragraph. For improving the reading experience of readers, one table should be keep a complete status (Table 1).

5. In the part of discussion, the authors of the paper need to do a more deep analysis in transferrin glycosylation how to influence iron metabolism of kidney disease.

Overall, the work is well-conducted.

The results of the study highlight the importance of glucose-free dialysate in controlling anemia exacerbation in PD patients with poorly controlled anemia, potentially providing a targeted therapeutic approach to improve patient prognosis, warrants further validation.

Comments on the Quality of English Language

I suggest only a minor revision of the English language for some grammatical errors and a check on the references.

Reviewer 2 Report

Comments and Suggestions for Authors

Dear Authors,
thank you for preparing this interesting manuscript entitled "Glycosylation and characterisation of human transferrin in an end stage kidney disease". In general, it is a well-written and easy-to-follow article with quite important results, however, I have two major concerns about this work:

1. Glycosylation is a very complex process that occurs in cells. The authors undertook the task of determining glycosylation and characterization of transferrin from a selected disease entity (end-stage kidney disease). This is a justified attempt, but the methods chosen for this study are not sufficient to confirm/deny the main research hypothesis - structural changes, in particular aberrant glycosylation, occur in transferrin from patients. In the professional literature (https://doi.org/10.1016/j.cbpa.2009.07.022; https://doi.org/10.1002/9780470054581.eib349), more specific methods (mainly those coupled with mass spectrometry) are used to analyze differences in the glycosylation (especially aberrant glycosylation) of a given protein, thanks to which it is possible to determine, for example, the hypothetical structure of glycans present in a given protein. Protein glycosylation may influence the structure of a protein, but it is not always the case, whereas the proper function of such protein is typically disturbed. Thus, in my opinion, this manuscript requires either additional investigation of the glycan structure (by other approved methods) or changing the emphasis of the text from the occurrence of altered glycosylation in this protein to hypothesizing this phenomenon. Lectin arrays are based on specific interactions of lectins with sugar residues but are not a reliable indicator of altered glycosylation.
2. In terms of measuring Tf interaction with TfR, I have the same opinion. The applied method (immunocytochemistry) is qualitative, not quantitative. In this case, I would suggest using a method (like MST, SPR, QCM) with which the Authors could measure some kinetic parameters of the occurring interaction e.g., affinity (Kd). If not this part of the manuscript should also be rewritten.

Some minor suggestions for this manuscript are:

1. the most common notation in the literature is "end-stage kidney disease"
2. please correct the notation of values and percentages to unseparated form (e.g. lines 41, 53, 107...) - for example, in lines 125-127 they are not separated
3. Please transfer Table 1 and its caption to the new page
4. In chapter 2.2 (Table 2), did the Authors use commercial transferrin or patient-derived transferrin? This is quite important as the patient-derived proteins can give aberrant results.
5. In Table 2, please correct the Latin names of sources to be written in italics
6. line 123, please correct spelling "differences"
7. methods description (Chapter 2.2), please add information upon the sample storage. (or maybe it was measured directly after isolation?)
8. line 292, please correct the entry (1×105 cells/well)

Round 2

Reviewer 2 Report

Comments and Suggestions for Authors

Dear Authors,

thank you for preparing the revised version of the manuscript.

Although I still believe that in this study applying some additional methods would be beneficial for the manuscript (for glycan structure evaluation and kinetic parameters quantification), I understand that this would be a time-consuming process and perhaps the samples are not available anymore.

Nevertheless, I am quite happy with the improved version of the manuscript. It is good that the Authors at least mention these methods and the occurred problems. But please consider them for future studies.
With kind regards